# Video Anomaly Detection via Progressive Learning of Multiple Proxy Tasks

Menghao Zhang
State Key Laboratory of Networking
and Switching Technology, Beijing
University of Posts and
Telecommunications
Beijing, China
E-BYTE Technology Co., Ltd.
Beijing, China
zhangmenghao@bupt.edu.cn

Jingyu Wang*
State Key Laboratory of Networking
and Switching Technology, Beijing
University of Posts and
Telecommunications
Beijing, China
E-BYTE Technology Co., Ltd.
Beijing, China
wangjingyu@bupt.edu.cn

Qi Qi
State Key Laboratory of Networking
and Switching Technology, Beijing
University of Posts and
Telecommunications
Beijing, China
E-BYTE Technology Co., Ltd.
Beijing, China
qiqi8266@bupt.edu.cn

Pengfei Ren
Beijing University of Posts and
Telecommunications
Beijing, China
rpf@bupt.edu.cn

Haifeng Sun
Beijing University of Posts and
Telecommunications
Beijing, China
hfsun@bupt.edu.cn

Zirui Zhuang*
Beijing University of Posts and
Telecommunications
Beijing, China
zhuangzirui@bupt.edu.cn

Huazheng Wang
Beijing University of Posts and
Telecommunications
Beijing, China
wanghz@bupt.edu.cn

Lei Zhang
Beijing University of Posts and
Telecommunications
Beijing, China
zhanglei@bupt.edu.cn

Jianxin Liao
Beijing University of Posts and
Telecommunications
Beijing, China
liaojx@bupt.edu.cn

## Abstract

Learning multiple proxy tasks is a popular training strategy in semi-supervised video anomaly detection. However, the traditional method of learning multiple proxy tasks simultaneously is prone to suboptimal solutions, and simply executing multiple proxy tasks sequentially cannot ensure continuous performance improvement. In this paper, we thoroughly investigate the impact of task composition and training order on performance enhancement. We find that ensuring continuous performance improvement in multi-task learning requires different but continuous optimization objectives in different training phases. To this end, a training strategy based on progressive learning is proposed to enhance the multi-task learning in VAD. The learning objectives of the model in previous phases contribute to the training in subsequent phases. Specifically, we decompose video anomaly detection into three phases: perception, comprehension, and inference, continuously refining the learning objectives to enhance model performance. In the three phases, we perform the visual task, the semantic task and the open-set task
in turn to train the model. The model learns different levels of features and focuses on different types of anomalies in different phases. Extensive experiments demonstrate the effectiveness of our method, highlighting that the benefits derived from the progressive learning transcend specific proxy tasks.

## CCS Concepts

• **Computing methodologies → Scene anomaly detection**; *Activity recognition and understanding*.

## Keywords

video anomaly detection, progressive learning, multi-task learning

### ACM Reference Format:

Menghao Zhang, Jingyu Wang, Qi Qi, Pengfei Ren, Haifeng Sun, Zirui Zhuang, Huazheng Wang, Lei Zhang, and Jianxin Liao. 2024. Video Anomaly Detection via Progressive Learning of Multiple Proxy Tasks. In *Proceedings of the 32nd ACM International Conference on Multimedia (MM '24), October 28-November 1, 2024, Melbourne, VIC, Australia*. ACM, New York, NY, USA, 10 pages. https://doi.org/10.1145/3664647.3680871

*Corresponding authors

## 1 Introduction

Video Anomaly Detection (VAD) is an essential task in multimedia interpretation, referring to the detection of unexpected events in videos. The primary challenge of VAD is the sparsity of abnormal samples, limiting the possibility of training efficient detectors in a fully supervised manner. Consequently, previous research in VAD predominantly falls into two categories: weakly supervised methods [8, 30, 41, 44, 48, 56, 60] that learns with video-level annotations, and semi-supervised methods that only learn from normal frames

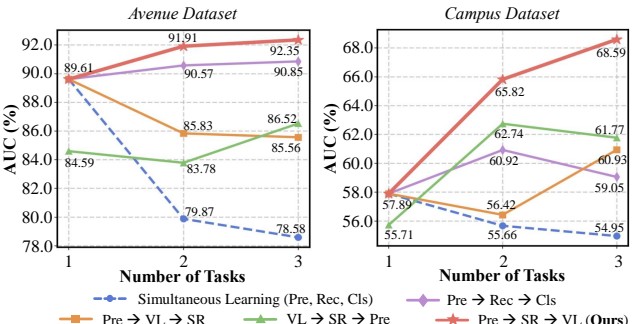

Figure 1: AUC (%) performances of anomaly detection when learning multiple pretext tasks on the Avenue [27] and Campus [3] datasets. Dashed lines indicate learning multiple proxy tasks simultaneously. Solid lines indicate training different tasks at different phases. "Pre", "Rec", and "Cls" represent the visual proxy tasks "prediction", "reconstruction", and "classification", respectively. "SR" represents the semantic proxy task "semantic reconstruction". "VL" represents the open-set proxy task "vision-language".

[5, 25, 26, 43, 50]. Our work focuses on the latter, which aligns with real-world settings.

Semi-supervised methods typically employ single or multiple proxy tasks to model normal patterns, with samples deviating from the established model labeled as anomalies. The single-proxy task method [5, 24, 35, 47, 50] is simple and effective when dealing with data involving a single scenario. In order to make the detector more adaptable to the complex and dynamic scenarios of the real world, some works [10, 11, 25] attempt to introduce multi-task learning to guide the model in acquiring distinct features for different types of anomalies. However, recent research [39] indicates that optimizing multiple proxy tasks simultaneously may not result in the expected performance enhancement. As shown by the dashed lines in Fig. 1, it could potentially be inferior to the performance of performing the single proxy task.

To ensure that multi-task learning achieves the desired performance improvements for VAD, a straightforward approach is to learn multiple proxy tasks sequentially, rather than attempting to learning them simultaneously. As depicted in Fig. 1, the sequential learning strategy yields superior performance gains compared to both the single-proxy task design and the simultaneous learning of multiple tasks, across datasets representing both single and complex scenarios (the Avenue [27] and Campus [3] datasets, respectively). However, not every set of proxy tasks and any training order can guarantee continuous performance improvement. Therefore, we ask the following questions in this paper: *for the training of video anomaly detector, (i) which proxy tasks are necessary, and (ii) what sequence of training is reasonable and effective.*

In fact, proxy tasks and training sequence are not completely decoupled. To ensure effective performance enhancement, proxy tasks in different training phases should entail distinct optimization objectives. To maintain the continuity of performance improvement, the optimization objectives across different training phases should be be consecutive. In a word, the selection of proxy tasks

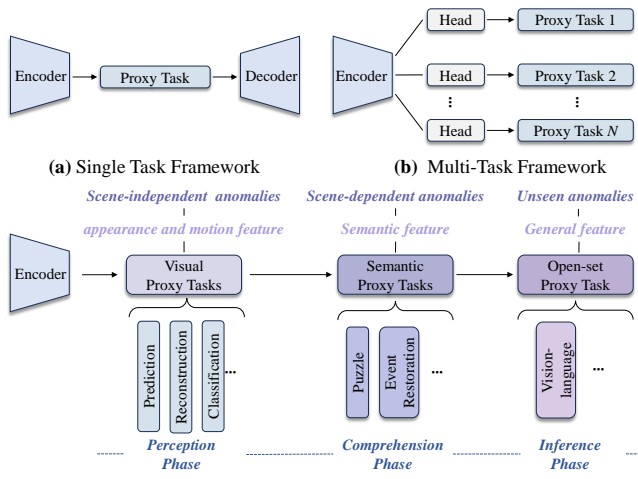

Figure 2: Comparison on the various designs of VAD framework. (a) Most solutions perform a single proxy task to train the detector. (b) Multi-task learning solutions share a backbone with divided task heads, learning multiple proxy tasks simultaneously. (c) Our proposed progressive learning design is a multi-phase training strategy for the purpose of anomaly detection, continuously refining learning objectives to tackle more challenging tasks.

and the training sequence should complement each other, different training phases require a series of consecutive optimization objectives. Therefore, the ideal approach is progressive learning rather than simply sequential learning, i.e., the learning objectives of the preceding phase contributing to the training for subsequent tasks.

Inspired by the human cognitive process, we decompose video anomaly detection into three phases: *perception, comprehension, and inference.* As shown in Fig. 2, our progressive learning design is a multi-phase training strategy for the purpose of anomaly detection, continuously refining learning objectives to tackle more challenging tasks. In the perception phase, the model aims to learn low-level visual information, laying the foundation for understanding the scene. In the comprehension phase, the model aims to learn semantic features in the scene, understanding the activities within the scene. In the inference phase, the model utilizes the learned knowledge and context to detect anomalies. Additionally, the decomposition approach enables the model to learn different levels of features at different phases, focusing on different types of anomalies. In the perception phase, the model focuses on obvious scene-independent anomalies (such as fire, smoke). In the comprehension phase, it combines scene information to address scene-dependent anomalies. In the inference phase, the model learns general features to address unseen anomalies encountered during training.

Correspondingly, **Answer to Question 1:** The training of detector requires three types of proxy tasks: the visual proxy task for perceiving pixel features, the semantic proxy task for comprehending semantic features, and the open-set proxy task for learning general features for inference. **Answer to Question 2:** The training sequence begins with visual proxy tasks, followed by semantic

proxy tasks, and concludes with open-set proxy tasks. To this end, we adopt the widely-used frame prediction task as the visual proxy task in the perception phase. As for the semantic proxy tasks in the comprehension, we propose a simple yet effective semantic reconstruction task leveraging the semantic consistency of context. During the inference phase, we adhere to previous work [26] by training the model with virtual data [1] to enhance its ability to detect unseen anomalies.

We conduct experiments on four challenging datasets (Avenue [27], ShanghaiTech [28], UCF-Crime [41] and Campus [3]). The experiments show that our proposed method outperforms the State-Of-The-Art (SOTA) methods and enhances the performance of multi-task learning. Furthermore, extensive experiments with widely-used proxy tasks demonstrate that the benefits derived from progressive learning extend beyond specific proxy tasks. In summary, our contributions can be summarized as follows:

- A training strategy based on progressive learning is proposed to enhance the efficacy of multi-task learning in VAD. We demonstrate the necessity of reasonable task composition and training sequences, rather than simply learning multiple tasks simultaneously or sequentially.
- Video anomaly detection is decomposed into three phases: perception, comprehension and inference. The model focuses on different types of anomalies in different phases.
- Extensive experiments demonstrate the effectiveness of our method, highlighting that the benefits derived from the progressive learning transcend specific proxy tasks.

## 2 Related Work

### 2.1 Video Anomaly Detection

Previous semi-supervised anomaly detection methods can mainly be classified into two categories: single-proxy-task [4, 17, 19, 35, 47] and multi-proxy-task methods [10, 11, 25]. Specifically, the term "semi-supervised" refers to methods that only use normal data during training. Most semi-supervised methods train models to perform a single proxy task on normal samples and assume that the model cannot perform the proxy task well on anomalous samples. The reconstruction task [12, 15, 29, 33, 58] to reconstruct the current frame and the prediction task [5, 23, 46, 53] to predict future frames are two commonly used proxy tasks, where frames with large reconstruction or prediction errors are recognized as anomalous. In addition, some work synthesize virtual anomaly data [1, 26] or proposes more complex proxy tasks to learn semantic features of normal patterns, including event completion [52], jigsaw puzzles [45], event restoration [50], and causal consistency [24].

While the single-proxy-task design yields promising results on datasets featuring a single scene, its performance falls short of expectations in real-world scenarios characterized by diverse scenes. This discrepancy arises from inconsistencies between the tasks during training (such as frame reconstruction or frame prediction) and those encountered during testing (such as anomaly detection). Therefore, recent works [9–11, 25, 31, 43] utilize multiple proxy tasks to train the model. For instance, Some works introduce information from other modalities, such as optical flow [25, 31] and skeletal information [43], where proxy tasks for each modality are jointly used as learning objectives. Some scene-aware methods

[2, 43, 55] separately design proxy tasks for scenes and objects to learn multi-level features. Meanwhile, other research [9–11, 43] designs an shared backbone with multiple task heads to simultaneously learn multiple proxy tasks targeting different anomalies.

However, the exploration of leveraging multiple proxy tasks to improve anomaly detection performance has not been thoroughly investigated in these studies. The challenge lies in the fact that simultaneously learning multiple proxy tasks can often lead to convergence to suboptimal solutions. Different from these methods, we extensively explore the impact of task composition and training sequence on multi-task learning in VAD. We find that in order to ensure continuous performance improvement, different but consecutive optimization objectives in different training phases are necessary. Based on this principle, we propose optimizing multiple proxy tasks through the progressive learning strategy. In addition, the gains from progressive learning are not limited to specific proxy tasks and their performance.

### 2.2 Multi-task Learning

Multitask learning [18, 20, 21, 32] is attracting increasing attention in the field of deep learning. Some methods achieve simultaneous learning of multiple tasks by designing complex network architectures [16] and optimization strategies [14, 38]. These methods are supervised and aim to improve the overall performance of multiple tasks. Unlike these methods, we aim for continuous performance improvement of the model as tasks are added. To achieve this, we perform different tasks at different training phases, progressively enhancing the model's performance. Progressive learning offers different but continuous optimization objectives at different training phases, with learning from previous phases aiding training in subsequent phases.

In video anomaly detection, prior work [39] attempts to sequentially learn multiple proxy tasks to ensure performance enhancement from multi-task learning. They arrange the training order based on the difficulty of tasks. However, continuous performance improvement is not available for sequential learning in all combination of tasks. Instead of executing the proxy tasks sequentially, we implement them in phases at various levels to ensure that the objectives are distinct yet consecutive. Specifically, inspired by the cognitive principle of understanding from the surface to the core, we decompose video anomaly detection into perception, comprehension, and reasoning phases. The model learns different proxy tasks at each phase to progressively acquire features at different levels. This decomposition allows the model to focus on different types of anomalies at different phases.

## 3 Method

### 3.1 Overview

Our proposed video anomaly detection framework progressively learns multiple proxy tasks to ensure that the model performance continuously improves as more tasks are learned. We aim to design a series of different and consecutive optimization objectives in different training phases, so that the feature learned in the previous training task contributes to the subsequent training. Specifically, we decompose video anomaly detection into three consecutive phases: perception, comprehension, and inference.

**Table 1: Introduction to the proposed progressive learning: In each column, from left to right, are the different phases and the corresponding types of target anomalies, the level of features to be learned, and the types of proxy tasks**

| Phase | Target | Learned Feature | Proxy Tasks |
|---|---|---|---|
| Percep­tion | Scene-independent Anomalies | **Pixel-Level** *color* *shape* *optical flow* | **Visual Tasks** *classification* *reconstruction* *prediction* |
| Comprehen­sion | Scene-dependent Anomalies | **Semantic-Level** *spatial feature* *temporal feature* *scene feature* | **Semantic Tasks** *completion* *restoration* *jigsaw puzzle* |
| Inference | Unseen Anomalies | **General Feature** | **Open-Set Tasks** *virtual data* *pseudo labels* *vision-language* |

The overall design of our proposed progressive learning is described in Table 1. For a normal video clip $V$: In the perception phase, we perform visual proxy tasks to learn pixel features of normal patterns, such as color, shape, and optical flow. In the comprehension phase, we perform semantic proxy tasks to learn semantic features of normal patterns, such as spatial features, temporal features, and scene semantics. In the inference phase, we perform the open-set proxy task to learn more discriminative general features. The open-set approach in VAD refers to the use of synthetic virtual anomalies or the acquisition of pseudo-anomalies to supplement the supervision instead of training with only normal data. In the process of progressive learning, the pixel features learned by the model in the perception phase provide the basis for comprehension, while the semantic features learned in the comprehension phase help the model to learn genreal features.

The network comprises three task heads and an encoder $F$. All task heads share the same backbone encoder $F$ to leverage knowledge learned from the previous phases, and all proxy tasks within the same phase share the same task head $H$. Next, we illustrate our progressive learning strategy with examples of frame prediction , semantic reconstruction, and virtual data-based classification tasks.

## 3.2 Data Acquisition

Pre-trained video parsing networks are widely used in VAD methods [10, 11, 25, 43] to extract different visual information. In this work, we utilize pre-trained networks to organize the training data.

In the perception phase, we perform the prediction task to learn pixel features. Given a video clip $V$, we use YOLOv3 [34] to extract all foreground objects in the video frame. Each object is identified by an RoI bounding box. For each RoI, we construct a spatio-temporal cube (STC) that contains not only the objects in the frame, but also the contents of the same bounding box from the previous $t$ frames, where $t = 4$. The width and height of the STCs are resized to 48.

In the comprehension phase, we perform the semantic reconstruction task to learn semantic features. In order to simultaneously learn the semantic features of the scene, we extracted the scene features of the background. For each frame in the clip $V$, we use DeepLabV3+ [6] to generate segmentation maps while masking

foreground object categories. We then perform max-pooling, reshaping, averaging, and $l_2$ normalization on all segmentation maps to obtain a scene feature $f^{scene} \in \mathbb{R}^{D_B}$, where $D_B$ depends on the size of the video frame.

In the inference phase, we train the classification task with virtual data to learn highly discriminative features. Following previous work [26], we use the virtual dataset UBnormal [1]. Unlike the previous two phases, the initial input to the model is video frames rather than object-level STCs.

## 3.3 Network Architecture

We employ the U-net [39] with deleted skip connections as the backbone network $F$ and use it as a shared base for the proxy tasks. Each training phase introduces into its top a task head, denoted $H_{per}$, $H_{comp}$, and $H_{inf}$, which are used to process the feature embeddings extracted from the backbone network. All task heads contain spatio-temporal averaging pooling layers to ensure that the dimensionality of the deep feature embeddings remains consistent, even though the temporal lengths may be different.

In perception phase and comprehension phase, we use task heads to embed the video features extracted by the backbone network into the low-dimensional features to get the representation. To be specific, the pixel feature representation and the semantic feature representation are obtained in the prediction task and the semantic reconstruction task, respectively. In inference phase, we use fully connected layers as classifiers at the end of $H_{inf}$ for performing classification task with virtual data. It should be noted that the parameters of the backbone are trainable. Since the learning objectives of previous phases contribute to subsequent training, retaining knowledge from earlier proxy tasks is advantageous. Although there are three parallel task heads in our architecture, only one task head performs the corresponding proxy task to optimize the shared backbone in each training phase.

## 3.4 Training Phases and Proxy Tasks

**Perception Phase.** We perform the frame prediction task as the visual proxy task to learn pixel-level features. We train our model to predict future frames based on a clip $V$ consisting of $2T + 1$ consecutive frames. In order to fully learn the pixel features of the normal model and provide a basis for subsequent learning. We train the model to predict future frames in two directions, forward and backward.

Formally, We construct a forward clip $\overrightarrow{V} = [X_1, X_2, \ldots, X_T]$ and a backward clip $\overleftarrow{V} = [X_{2T+1}, X_{2T}, \ldots, X_{T+2}]$ as the initial inputs. X does not represent the video frames in the clip, but the combination of all the spatio-temporal cube in each frame. The object-level $X_{T+1}$ of the middle frame is the prediction target. We use the shared backbone $F$ to encode these two clips first and use the prediction header $H_{per}$ to predict $X_{T+1}$. The final prediction $\hat{X}$ is calculated as:

$$\hat{X} = \frac{H_{per}(F(\overrightarrow{V})) + H_{per}(F(\overleftarrow{V}))}{2}. \tag{1}$$

where $(\cdot)$ represents the process of feature processing by the network. We use the $l_2$ normalization to evaluate the difference between the prediction and the ground truth $X_{T+1}$. The prediction

loss $\mathcal{L}_{pre}$ is given as:

$$\mathcal{L}_{\text{pre}} = \left\| \hat{X} - X_{T+1} \right\|_2 . \tag{2}$$

To generate smooth prediction frames, we also utilize gradient loss. The gradient loss $\mathcal{L}_{gd}$ is calculated as:

$$\mathcal{L}_{\text{gd}} = \left\| \nabla \hat{X} - \nabla X_{T+1} \right\|_1 , \tag{3}$$

where $\nabla$ and $|\cdot|_1$ represent gradient computation and $l_1$ normalization, respectively. The prediction loss $\mathcal{L}_{\text{pre}}$ and gradient loss $\mathcal{L}_{\text{gd}}$ are the optimization objectives in this phase.

**Comprehension Phase.** In order to further learn the higher-level semantic features of the video clip $V$, we introduce a semantic reconstruction task as the semantic proxy tasks. We design a contrastive learning mechanism to perform semantic reconstruction task based on the following facts. In the context of video frames, frame $I^{anc}$ should exhibit semantic consistency with its neighboring frames due to their temporal continuity, as opposed to disjoint frames.

To perform the proposed semantic reconstruction tasks, we select positive video clips and negative clips containing only one anchor frame $I^{anc}$, respectively. The negative clip contains $T$ continuous frames, while the positive clip has a length of $T-1$. The reason for this phenomenon is that we use masking to select the anchor frame. Given an initial continuous clip of length $T$, we uniformly select a frame from the interval $[2, T-1]$ as the anchor frame $I^{anc}$. And the leading and alternate of the missing frames are concatenated together to form positive video clip $V^d$.

Accordingly, we learn the motion information in the video by triplet loss [37] taking the discontinuous video clips $v_i^d$ as the anchor point, the masked frames $I^m$ as the positive samples, and the continuous video clips $v_i^c$ as the negative samples. Defining the cosine similarity computation operation as $sim(\cdot, \cdot)$, the semantic reconstruction loss is denoted as:

$$\mathcal{L}_{SR} = \frac{1}{N} \sum_{i=1}^{N} \max(0, \gamma - (p_i^+ - p_i^-)) \tag{4}$$

$$p_i^+ = \text{sim}\left(H_{com}([F(V_i^d), f^{scene}], [F(I_i), f^{scene}])\right), \tag{5}$$

$$p_i^- = \text{sim}\left(H_{com}([(V_i^d), f^{scene}], [F(V_i^c), f^{scene}])\right), \tag{6}$$

where $p_i^+$ and $p_i^-$ denote the similarity between positive sample pairs and negative sample pairs in the triplet loss, respectively. $[\cdot]$ denotes the concatenation. The semantic reconstruction loss $\mathcal{L}_{SR}$ is the optimization objective in comprehension phase.

**Inference Phase.** With the help of virtual data containing anomalous data, we use the classification task as the open-set proxy task. With $N$ samples in a training batch, cross-entropy loss $\mathcal{L}_{CE}$ is used to optimize the model, and the optimization objectives in this phase is $\mathcal{L}_{cls}$:

$$\mathcal{L}_{cls} = \frac{1}{N} \sum_{i=1}^{N} \left( \mathcal{L}_{CE} \left( P_1 \left( F \left( v_i^{anom} \right) \right)_1 + P_1 \left( F \left( v_i^{nor} \right) \right)_0 \right) \right), \tag{7}$$

where $(\cdot)$ represents the process of feature processing by the network, $F$ and $H_{inf}$ represent the shared backbone network and the task head of inference phase, respectively, and $v_i^{nor}$ and $v_i^{anom}$ represent the $i_{th}$ normal and anomalous frame in the virtual data, respectively.

**Table 2: Ablation experiments on the multi-task learning. We report the AUC (%) scores on ShanghaiTech and Campus datasets. 'FP', 'Rec', 'SR', 'Jig', and 'VL' stand for the proxy tasks of frame prediction, reconstruction, proposed semantic reconstruction, jigsaw puzzle, and vision-language, respectively. In addition, "Vir" stands for we use virtual data for classification training.**

| ID | FP | Rec | SR | Jig | VL | Vir | AUC | |
|----|----|----|----|----|----|----|--------|--------|
| | | | | | | | ShTech | Campus |
| 1 | - | - | - | ✓ | - | - | 76.5 | 62.6 |
| 2 | ✓ | - | - | - | ✓ | - | 70.1 | 61.9 |
| 3 | - | ✓ | - | - | ✓ | - | 71.3 | 61.1 |
| 4 | - | - | ✓ | - | - | ✓ | 78.8 | 65.2 |
| 5 | ✓ | ✓ | - | - | ✓ | - | 73.9 | 64.4 |
| 6 | ✓ | - | ✓ | - | - | ✓ | 79.0 | 67.2 |
| 7 | - | ✓ | ✓ | ✓ | - | - | 78.4 | 61.8 |
| 8 | ✓ | ✓ | - | ✓ | - | ✓ | **81.2** | **68.8** |
| 9 | ✓ | ✓ | ✓ | ✓ | ✓ | ✓ | 75.8 | 65.6 |

## 3.5 Anomaly Detection

During testing, we first calculate the prediction error $L_{pre}$ for each frame in clip $V$, calculated as:

$$\mathcal{L}_{pre}(V) = \left\| H_{\text{pre}}\left(F(V)\right) - X_{T+1} \right\|_2 . \tag{8}$$

We then assign the anomaly score of any frame in which no salient objects are detected to 0. For a frame with $m$ salient objects, the maximum anomaly score $S_{max}$ between these objects is:

$$\mathcal{S}_{\max} = \max\left\{\mathcal{L}_{pre}(V_1), \mathcal{L}_{pre}(V_2), \ldots, \mathcal{L}_{pre}(V_m)\right\} \tag{9}$$

Following [39], frame-level anomaly scores are further smoothed by a median filter with a window size of 17 to ensure temporal consistency of the video.

## 4 Experiments

## 4.1 Experimental settings

**Scene-Independent Anomaly Dataset** Avenue [27] contains 16 training videos and 21 test videos with 47 abnormal events, including running and throwing. The scenes in this dataset are single and the anomalies are related to human. ShanghaiTech [28] has 13 scenes with complex lighting conditions and different perspectives. In addition, the dataset includes anomalies caused by sudden movements, such as chasing and arguing. The different perspectives and the unfixed position of the camera lead to a large variation of both the object scale and the anomaly scale in the scene. UCF-Crime [41] comprises 13 anomaly types, spanning a total of 128 hours of video footage. The training set contains video-level labels, whereas the testing set includes frame-level labels.

**Scene-Dependent Anomaly Dataset.** Campus [3] is currently the most challenging dataset in its field with 43 scenes, 28 classes of anomalous events and 16 hours of videos. Especially, it contains scene-dependent anomalies, which means an normal event may be abnormal in another scene. Detecting scene-dependent anomalies requires the model to understand the scene and learn semantic features rather than overfit.

**Table 3: AUC (%) performance of models trained with different combinations of tasks using different learning sequences and models trained simultaneously with the same weights for all tasks on the ShanghaiTech and Campus datasets. The results indicate that our progressive learning approach achieves the maximum performance improvement when learning multiple auxiliary tasks, and the performance gains are not limited to specific tasks.**

| Dataset | Progressive | | | | | | Simultaneous |
|---|---|---|---|---|---|---|---|
| | Learning Order | | | AUC | | | AUC |
| | Phase 1 | Phase 2 | Phase 3 | Phase 1 | Phase 2 | Phase 3 | |
| ShanghaiTech | Pre | →SR | → Virtual | 73.1 | 79.2 | 83.8 | 79.0 |
| | Pre | →Virtual | →SR | 73.1 | 72.0 | 75.5 | 79.0 |
| | SR | →Pre | →Virtual | 76.1 | 75.5 | 76.8 | 79.0 |
| | Cls | →Virtual | →SR | 65.4 | 64.2 | 73.2 | 70.1 |
| | - | →Virtual | →SR | - | 68.8 | 76.2 | 72.1 |
| | Rec | →Jigsaw | →VL | 71.9 | 81.2 | 82.3 | 80.2 |
| | Rec | →Completion | →VL | 71.9 | 79.8 | 81.9 | 79.5 |
| | Rec | →Completion | →Pseudo | 71.9 | 79.8 | 81.1 | 80.0 |
| | Pre → Rec | - | → Pseudo | 76.0 | 76.0 | 79.3 | 79.0 |
| | Pre → Rec | → SR | - | 76.0 | 85.0 | 85.0 | 82.3 |
| | Pre → Rec | → Jig | → Virtual | 76.0 | 85.9 | 88.6 | 79.2 |
| | Pre → Rec | → Jig → Completion | → Virtual | 76.0 | 86.1 | 88.8 | 78.1 |
| | Pre → Rec → Cls | → SR → Jig → Completion | → Virtual | 76.8 | 86.4 | 88.8 | 77.9 |
| Campus | Pre | →SR | → Virtual | 57.9 | 66.2 | 69.4 | 67.2 |
| | Pre | →Virtual | →SR | 57.9 | 57.0 | 68.1 | 67.2 |
| | SR | →Pre | →Virtual | 65.4 | 64.8 | 66.1 | 67.2 |
| | Cls | →Virtual | →SR | 54.2 | 53.9 | 62.8 | 62.0 |
| | - | →Virtual | →SR | - | 58.1 | 64.3 | 64.2 |
| | Rec | →Jigsaw | →VL | 55.1 | 71.2 | 72.1 | 68.0 |
| | Rec | →Completion | →VL | 55.1 | 67.9 | 70.6 | 67.2 |
| | Rec | →Completion | →Pseudo | 55.1 | 67.9 | 71.1 | 67.8 |
| | Pre → Rec | - | → Pseudo | 58.2 | 58.2 | 60.6 | 61.3 |
| | Pre → Rec | → SR | - | 58.2 | 69.9 | 69.9 | 66.1 |
| | Pre → Rec | → Jig | → Virtual | 58.2 | 71.2 | 73.3 | 70.2 |
| | Pre → Rec | → Jig → Completion | → Virtual | 58.2 | 72.3 | 75.1 | 66.9 |
| | Pre → Rec → Cls | → SR → Jig → Completion | → Virtual | 59.1 | 74.5 | 75.8 | 66.8 |

**Evaluation Metrics.** The area under the ROC curve (AUC) serves as a commonly used metric for evaluation and comparison. A higher AUC score indicates a better anomaly detection capability.

## 4.2 Implementation Details

We train and evaluate our method with an NVIDIA RTX 3090 GPU. In the training phase, we resize the resolution of all input video clips to $256 \times 256$ pixels, while the values of the pixels in all frames are normalized to the range [0, 1]. For the pre-training of the three proxy tasks, we utilize AdamW as the optimizer while the length of the continuous video clips is set to $T = 9$ frames. The initial learning rate is set to 0.0003 and is gradually decayed following the scheme of cosine annealing. In our reported experimental performance, the shared backbone network $F$ consists of a convolutional layer and three convolutional blocks. The structure and all experiments are implemented in PyTorch.

## 4.3 Ablation Study

Given the diversity of perspectives and scenes within the ShanghaiTech and Campus datasets, we conduct comprehensive ablation experiments on both datasets.

In addition, to demonstrate that progressive learning is not limited to specific tasks, we also conduct experiments with other proxy tasks besides the three mentioned above. For the visual proxy task, we select the reconstruction [39] and classification [39] tasks. For the semantic proxy task, we select the event completion [52] and puzzle task [45]. For the open-set proxy task, we select the background-agnostic [11] of synthesizing pseudo anomalies and the Vision-Language task [49] of generating novel anomalies using pre-trained multi-modal model.

**Preliminary Experiments on Multi-Tasks Learning.** As shown in Fig. 1, simultaneous multi-task learning in VAD may lead to model convergence to the sub-optimal point. We perform experiments using more proxy tasks and the results are reported in Table 2. According to the experimental results, executing all proxy tasks does not yield the best performance (ID 8, 9). In fact, its performance is inferior to executing a single proxy task (ID 1). The experiments for IDs 6, 8 achieve superior performance with fewer proxy tasks. Notably, the experiment with ID 9 showed improved performance on the Campus dataset compared to experiments with IDs 1 and 7. However, the performance on the ShanghaiTech dataset decreased, indicating that the model's boost in performance on one task came

at the expense of performance on other tasks. In addition, executing visual, semantic and open-set proxy tasks simultaneously will all yield better performance (ID 6, 8).

**Ablation Study on Proxy Tasks Composition.** As shown in Table 3, we conduct ablation experiments with different combinations of tasks. Models in which all three types of proxy tasks are performed tend to obtain better performance when the training order is consistent. The performance of the model is significantly lower when a phase of the proxy task is missing. This improvement is not limited to specific tasks. When there are more semantic proxy tasks, the model achieves better performance on the Campus dataset, which is attributed to the fact that the Campus dataset contains mainly scene-dependent anomalies.

**Ablation Study on Learning Sequences.** As shown in Table 3, only the models that followed this training order of "Visual proxy tasks → Semantic proxy tasks → Open-set proxy tasks" are continuously improving in performance. This is because progressive learning provides different but consecutive learning objectives. The learning objectives of the previous task help to learn the subsequent learning objectives. When the training sequence is not progressive learning, the model's performance will show a decrease rather than a continuous increase during training.

**Sensitivity to the Number of Proxy Tasks in Each Phase.** In Section 3, we perform only one proxy task at each phase in order to succinctly present the training strategy for progressive learning. In fact, the performance of the model improves as more tasks are performed at each phase (there is an edge effect). As shown in Table 3, following the learning sequence of progressive learning, the performance of the model continuously improves with the increase of tasks. However, there is a marginal effect of performance growth with each phase of the task.

**Table 4: Ablation experiments for simultaneous learning of reweighted multiple proxy tasks. We report AUC (%) on ShanghaiTech and Campus datasets. The best performing results are marked in bold.**

| $\omega_1 : \omega_2 : \omega_3$ | AUC | |
| --- | --- | --- |
| | ShanghaiTech | Campus |
| 6:3:1 | 70.3 | 60.5 |
| 4:2:1 | 72.9 | 62.1 |
| 1:2:1 | 82.2 | 66.9 |
| 1:3:1 | 81.5 | 68.0 |
| 1:2:4 | 78.4 | 65.3 |
| 1:3:6 | 77.5 | 65.0 |
| Progressive Learning | **83.8** | **69.4** |

**Ablation Study on Re-weighting.** We conducted the following experiments to verify that our progressive learning approach cannot be replaced by reweighting of multiple proxy tasks. We assign different weights to each stage of the training tasks then learn them simultaneously. Higher weights represent that in learning the model pays more attention to the learning objective of this training task. The experimental results are shown in Table 4. The combination of proxy tasks in this experiment is: prediction → semantic reconstruction → virtual data.

**Table 5: Ablation Experiments for Each Phase. We report AUC (%) on Avenue, Campus, and Ubnormal datasets.**

| Phase | Avenue | Campus | Ubnormal |
| --- | --- | --- | --- |
| Per | 86.9 | 57.9 | 60.1 |
| Und | 85.8 | 64.3 | 62.1 |
| Inf | 83.2 | 58.1 | 66.5 |
| Per + Und | 92.0 | 66.2 | 63.9 |
| Per + Inf | 92.5 | 57.0 | 67.6 |
| Und + Inf | 89.2 | 65.5 | 69.2 |
| Per + Und + Inf | 93.6 | 69.4 | 71.8 |

**Ablation Experiments on Each Training Phase.** We conduct ablation experiments on three datasets, containing different types of anomalies, to show the effectiveness of each phase (Avenue dataset [27] for scene-independent anomalies, Campus dataset[3] for scene-dependent anomalies, UBnormal dataset [1] for unseen anomalies). The proxy tasks we perform in each of the three phases are frame prediction, semantic reconstruction and applying virtual data for training. Note that the virtual data we use for training is the training set of UBnormal to avoid the problem of data leakage. As shown in Table 5, after training on a particular phase, the model achieves significant improvements on the corresponding dataset.

## 4.4 Comparisons with State-Of-The-Arts

We compare the proposed framework with SOTA methods with AUC(%). It is worth noting that only a few works [1, 11, 26] in the current literature utilize virtual data for feature learning (marked with ∗ in Table 6). To ensure fair comparison, we report the performance of models trained with and without virtual data (i.e., whether open-set proxy tasks are performed during training). Furthermore, to fully demonstrate the effectiveness of progressive learning, we train models using two sets of proxy tasks and report performance accordingly. These sets consist of: a baseline version $model_{base}$ trained with simple direct proxy tasks (prediction, semantic reconstruction, and virtual data-based classification), and a sota version $model_{sota}$ trained with multiple widely-used proxy tasks (prediction, reconstruction, jigsaw puzzle, and virtual data-based classification).

Formally, the training sequence for $model^*_{base}$ is: prediction → semantic reconstruction → virtual data. The training sequence for $model^*_{sota}$ is: prediction → reconstruction → jigsaw → virtual data.

**Results on Avenue.** As shown in Table 6, our method achieved the highest AUC scores, obtaining a AUC of 94.5%. With the simple proxy tasks, our method still scored the best performance with 93.6% AUC scores.

**Results on ShanghaiTech.** Our proposed progressive learning method yields a AUC of 88.6% on the ShanghaiTech dataset. With the virtual data [26], we improve the AUC by 1.9% and by 3.8% when performing widely-used proxy tasks instead of simple tasks. With the simple proxy tasks, our method still achieves an AUC score of 83.8%, which is superior to most methods. In comparison with the method [39] sequentially learning multiple proxy tasks, our method achieves a significant improvement of 10.0%. The huge improvement over sequential learning and methods that learn

**Table 6: Comparison with SOTA methods of the AUC(%) on Avenue and ShanghaiTech datasets. The best performing results are marked in bold.**

| year | Method | Task | AUC | |
|---|---|---|---|---|
| | | | Avenue | ShTech |
| Before 2021 | Liu *et al.* [23] | single | 85.1 | 72.8 |
| | Gong *et al.* [13] | multiple | 83.8 | 71.2 |
| | Ionescu *et al.* [19] | single | 87.4 | 78.7 |
| | Park *et al.* [33] | multiple | 88.5 | 70.5 |
| | Liu *et al.* [25] | multiple | 91.1 | 76.2 |
| | Lv *et al.* [29] | multiple | 89.5 | 73.8 |
| | Georgescu *et al.* [10] | multiple | 91.5 | 82.4 |
| | Georgescu *et al.* [11]* | multiple | 92.3 | 82.7 |
| 2022 | Wang *et al.* [46] | single | 88.3 | 76.6 |
| | Wang *et al.* [45] | single | 92.2 | 84.3 |
| | Zaheer *et al.* [54] | multiple | 74.2 | 79.6 |
| | Chen *et al.* [5] | multiple | 90.3 | 78.1 |
| | Zhong *et al.* [59] | multiple | 89.0 | 74.5 |
| | Cho *et al.* [7] | single | 88.0 | 76.3 |
| | Yang *et al.* [51] | single | 89.9 | 74.7 |
| | Ristea *et al.* [35] | multiple | 92.9 | 83.6 |
| | Acsintoae *et al.* [1]* | multiple | 93.0 | 83.7 |
| 2023 | Yang *et al.* [50] | single | 89.9 | 73.8 |
| | Cao *et al.* [3] | single | 86.8 | 79.2 |
| | Liu *et al.* [22] | single | 92.8 | 78.8 |
| | Singh *et al.* [40] | multiple | 86.0 | 76.6 |
| | Sun *et al.* [43] | multiple | 92.4 | 83.0 |
| | Sun *et al.* [42] | single | 91.5 | 78.6 |
| | Shi *et al.* [39] | multiple | 91.5 | 78.6 |
| | Liu *et al.* [26]* | single | 90.9 | 78.8 |
| | Liu *et al.* [26] * | multiple | 93.6 | 85.0 |
| 2024 | Zhang *et al.* [57]* | multiple | 93.2 | 86.2 |
| | Ristea *et al.* [36] * | multiple | 91.3 | 79.1 |
| | $model^*_{base}$(Ours) | multiple | 93.6 | 83.8 |
| | $model^*_{sota}$(**Ours**) | multiple | **94.5** | **88.6** |

**Table 7: Results of AUC(%) on UCF-Crime dataset. The best performing results are marked in bold.**

| Method | Reference | Task | AUC |
|---|---|---|---|
| Park *et al.* [33] | CVPR20 | multiple | 68.9 |
| Georgescu *et al.* [10] | CVPR21 | multiple | 74.6 |
| Wang *et al.* [46] | TNNLS22 | single | 72.9 |
| Sun *et al.* [43] | CVPR23 | multiple | 75.5 |
| $model^*_{base}$(Ours) | - | multiple | **79.9** |
| $model^*_{sota}$(**Ours**) | - | multiple | **83.2** |

multiple proxy tasks simultaneously demonstrates the benefits of progressive learning for model training.

**Results on UCF-Crime.** Due to the absence of published results (methods only learning from normal data) on the UCF-Crime dataset, we implement the code from the existing literature [10,

**Table 8: Results of the AUC(%) on Campus dataset. The best performing results are marked in bold.**

| Method | Reference | Task | AUC |
|---|---|---|---|
| Liu *et al.* [23] | CVPR18 | single | 57.9 |
| Gong *et al.* [13] | CVPR19 | multiple | 61.9 |
| Ionescu *et al.* [19] | CVPR19 | single | 59.3 |
| Park *et al.* [33] | CVPR20 | multiple | 62.5 |
| Liu *et al.* [25] | ICCV21 | multiple | 63.7 |
| Georgescu *et al.* [10] | CVPR21 | multiple | 65.9 |
| Wang *et al.* [46] | TNNLS22 | single | 61.9 |
| Wang *et al.* [45] | ECCV22 | single | 65.8 |
| Cao *et al.* [3] | CVPR23 | single | 68.2 |
| Zhang *et al.* [57] | CVPR24 | multiple | 70.1 |
| $model^*_{base}$(Ours) | - | multiple | 69.4 |
| $model^*_{sota}$(**Ours**) | - | multiple | **73.3** |

33, 43, 46]. As shown in Table 7, our proposed method achieves an improvement compared to the second best method by 4.4% when only learning simple tasks, and 7.7% when learning complex tasks. **Results on Campus.** As shown in Table 8, our method achieves the highest AUC score of 73.3%. Without the complex proxy tasks, our method still obtain an AUC of 69.4%, surpassing other multi-task methods [10, 13, 25, 33].

Compared to the Avenue dataset, which features a single scene, the latter three datasets contain a greater variety of scenes and anomaly types. The superior performance of progressive learning on the four datasets and the improvement on the latter three datasets demonstrate its effectiveness.

## 5 Conclusion

In this paper, we thoroughly investigate the impact of task composition and training sequence on the performance improvement of video anomaly detection in multi-task learning. To ensure sustained performance improvement as the number of tasks increases, we propose progressive learning of multiple proxy tasks. Progressive learning provides different but continuous optimization objectives at different phases of training, allowing knowledge learned in previous phases to contribute to subsequent training phases. Specifically, we decompose video anomaly detection into three training phases: perception, comprehension, and inference. Extensive experiments on four challenging datasets not only demonstrate the effectiveness of our proposed method but also show that the gains brought by progressive learning are not limited to specific tasks.

## Acknowledgments

This work was supported by the National Natural Science Foundation of China under Grants (62101064, 62171057,62201072, U23B2001, 62001054, 62071067), the Ministry of Education and China Mobile Joint Fund (MCM20200202, MCM20180101), the Beijing University of Posts and Telecommunications-China Mobile Research Institute Joint Innovation Center, the Project funded by China Postdoctoral Science Foundation (2023TQ0039), and the BUPT Excellent Ph.D. Students Foundation (Grant CX20241007).

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
