# OpenReview forum: "Video Anomaly Detection via Progressive Learning of Multiple Proxy Tasks"
_acmmm.org/ACMMM/2024/Conference — MM2024 Poster_

### Official Review · Reviewer_wzPA · 2024-05-02

**Rating:** 5
**Confidence:** 4

**Summary:**

The authors propose a method of progressively learning multiple proxy tasks to improve the trained model's performance on anomaly detection. Proxy tasks are classified into different categories so that the features of anomalies are learned from coarse to fine. In this way, it's easier to incorporate new proxy tasks for enhancing model's performance.

**Strengths:**

1. This paper thoroughly explores how to progressively learn multiple proxy tasks for improving the performance of anomaly detection. Each proxy task is categorized into different phases. In this way, the training objective is constantly evolving with the training process, yet stays consistent with previous phases.
2. The authors have conducted numerous experiments and results are convincing to show the effectiveness of the proposed method.

**Limitations:**

1. The main motivation/idea of the work is incremental as it is largely built upon the previous work. Different tasks are heuristically classified into each phase. It would be better if the authors could provide a more in-depth analysis of the mechanism of this sequential learning method.
Minor mistakes:
  1. In my opinion, semi-supervised learning refers to learning with data that is partially labeled and I would not classify methods that only learn from normal frames as semi-supervised ones. In fact, there are previous methods which term this setting as one-class-classification.
  2. Typos&Grammatical mistakes - an incomprehensive list:
    line 89, learning - learn;  line 99, "to to guide the model" , multiple "to"s; in table 3, there is "Avenue" in the dataset column；line 448 the loss term L_gd is not the same as L_grd in the equation (3). There are more mistakes/typos in the manuscript. I highly suggest that the authors should carefully revise the manuscript.

**Suitability:**

2

---

### Official Review · Reviewer_JvvT · 2024-05-19

**Rating:** 4
**Confidence:** 4

**Summary:**

The paper presents a progressive learning strategy for video anomaly detection (VAD), which enhances the efficacy of multi-task learning. The authors decompose VAD into three phases, i.e., perception, comprehension, and inference, to refine learning objectives progressively. In the perception phase, visual proxy tasks are performed to learn pixel-level features. The comprehension phase involves semantic proxy tasks to understand anomalies within the scene, while the inference phase uses open-set proxy tasks to detect anomalies. The model learns different levels of features and focuses on various anomaly types across phases. Extensive experiments on several datasets demonstrate the method's effectiveness.

**Strengths:**

-The idea of progressive learning of multiple proxy tasks is interesting and promising in video anomaly detection.

-The experiment results seem solid and sufficient.

-The writing of this paper is commendable.

**Limitations:**

-The motivation needs to be further clarified. What is the specific insight principle and intuition behind the progressive learning of multiple proxy tasks? The explanation of the human cognitive process is somewhat vague and simple in the Introduction section.

-The authors have mentioned that the designed learning phases can address different types of anomalies. It would be better to provide a further explanation of the proposed anomaly classification and explain its reasonableness. And if necessary, please provide more examples to help readers understand.

-What is the motivation and purpose of leveraging the semantic consistency of context to design semantics proposed in the Abstract section?

-In Section 3.5, why do the authors only use the prediction error from the perception phrase to generate anomaly scores? What about the proxy tasks which focus on different types of anomalies in the other two phrases when generating anomaly scores? How to obtain salient objects in Eq. (9)? What would be the experimental result if all objects were used?

-Will the proposed progressive learning method lead to a catastrophic forgetting problem?

-Some minor mistakes: the“Avenue Campus” in Table 3, and the lack of bold marks in Table 4.

**Suitability:**

2

---

### Official Review · Reviewer_Q76W · 2024-05-28

**Rating:** 4
**Confidence:** 3

**Summary:**

This paper introduces a progressive learning training approach for multi-task learning in video anomaly detection, highlighting the importance of task integration. The progressive learning method involves three phases: perception, comprehension, and inference. These phases are utilized to acquire low-level visual information, semantic features, and detect anomalies. Through extensive experiments and ablation studies on four datasets, the effectiveness of the proposed method is demonstrated.

**Strengths:**

- Novelty and technical correctness: The novelty of the paper is moderate, as it proposes a progressive learning method for multi-task learning, which considers the selection of proxy tasks and training sequences.
- Adequate evaluation: The paper conducts experiments on four widely used anomaly detection datasets.
- Clarity: The majority of the writing and logic are clear.
- Applications: This method is significant for video anomaly detection area.

**Limitations:**

1. Are all layers of the shared base network frozen or trainable during the three training phases? Why does continuous learning help prevent the network from converging to suboptimal solutions? Doesn't continuous learning result in the network retaining knowledge from the last proxy tasks?
2. It is difficult to understand how virtual data is utilized during the inference phase. Could you please provide a detailed explanation?
3. Could you explain the rationale behind the learning order presented in Table 3? It seems that not all combinations of proxy tasks were considered.
4. Why do the first and second lines of Table 3 show different AUCs (79.0 and 74.2) when training the same proxy tasks simultaneously?
5. Numerous writing errors need to be corrected, such as 'L_grd' in Equation 3, 'L_pre' on line 504, 'cls' in Table 3, and 'rediction' on line 741.

**Suitability:**

3

---

### Official Review · Reviewer_XefE · 2024-05-28

**Rating:** 3
**Confidence:** 4

**Summary:**

The paper addresses the challenge of optimizing multi-task learning for semi-supervised video anomaly detection.

**Strengths:**

Innovative Approach: The paper introduces a novel progressive learning strategy, which is a fresh perspective in the realm of multi-task learning for VAD.
Comprehensive Experiments: The extensive experiments conducted validate the effectiveness of the proposed method, demonstrating substantial performance improvements.

**Limitations:**

Lack of Comparative Baselines: The paper would benefit from a more thorough comparison with a wider range of baseline methods to robustly establish the superiority of the proposed approach.
Theoretical Justification: There is a need for a deeper theoretical explanation for why the progressive learning strategy outperforms traditional multi-task learning approaches.
Task-Specific Insights: The paper does not provide detailed insights into how the model handles different types of anomalies across the three phases, which could help in understanding the nuances of the proposed method.
Scalability Concerns: The scalability of the proposed method to other domains or larger datasets is not discussed, which limits the generalizability of the approach.
Ablation Studies: More ablation studies are required to isolate the impact of each phase and the specific tasks within those phases, ensuring that each component’s contribution to overall performance is clear.

**Suitability:**

2

---

### Official Review · Reviewer_pLfm · 2024-05-28

**Rating:** 4
**Confidence:** 2

**Summary:**

This paper investigates the task of video anomaly detection. It begins by empirically demonstrating that learning with multiple proxy tasks, either sequentially or simultaneously, does not consistently yield improvements. To address this, the training phase is decomposed into three stages, progressively learning features from the pixel level to the semantic level. The results show the superiority of the proposed method.

**Strengths:**

There are several strengths in this paper:

- Overall, the idea of this paper is easy to comprehend and technically sound.

- The evaluation provides substantial evidence demonstrating the effectiveness of the proposed method, which is convincing.

**Limitations:**

However, I still have the following concerns:

- The motivation of this paper is not sufficiently supported. It is unclear if the shortcomings of sequential learning and multi-task learning are consistent across different methods and how severe they are. Although I agree with the paper's statements intuitively, more empirical evidence is needed.

- The overall training paradigm is overly complex, involving many engineering challenges, making it less practical for real-world scenarios.

- In lines 268-269, it states, "Unlike these methods, we aim for continuous performance improvement of the model as tasks are added." If tasks are added sequentially, why not compare with continual learning methods rather than multi-task learning?

- The introduction of multi-task learning in the related works section is too brief. The authors emphasize their progressive strategy more, which is not helpful for the audience.

- Since there are multiple objectives, applying a multi-objective optimization strategy is straightforward. Thus, I recommend the authors provide some insights on this to further support the paper's motivation. However, this is not mandatory, and it is acceptable if the authors choose not to include it.

Based on the current strengths and limitations, I still lean towards a positive evaluation of this paper. As I am not an expert in video anomaly detection, my confidence is somewhat decreased, but I remain open to discussion.

**Suitability:**

3

---

### Official Review · Reviewer_VBxK · 2024-06-09

**Rating:** 3
**Confidence:** 4

**Summary:**

This study examines how task composition and the order of training impact the performance of video anomaly detection within a multi-task learning framework. The researchers propose a novel approach called progressive learning for multiple proxy tasks. This method provides distinct yet continuous optimization goals throughout various training stages, allowing knowledge from earlier phases to improve subsequent training phases. Extensive experiments are conducted to demonstrate the proposed method's effectiveness. The paper is well-structured, clearly written, and the experimental results convincingly support the method's efficacy.

**Strengths:**

This paper is well-organized and easy to follow.
The experiments are sufficient to demonstrate the efficacy of progressive learning using a multi-phase training strategy for video anomaly detection. The results show that continuously refining learning objectives helps tackle more challenging tasks. The motivation is straightforward, the network is well-designed, and each task is explained with clear motivation. Overall, the motivation is sufficient for an MM conference paper.

**Limitations:**

1. On page 6, line 626, executing all proxy tasks does not yield the best performance (ID 8, 9). Please provide more detailed reasons for this outcome.
2. On page 6, the paper discusses progressive learning for video anomaly detection within a multi-task learning framework. Are there any criteria to automatically combine the multi-learning objectives instead of manually crafting them, while still achieving performance gains? Additionally, the paper states, "the performance of the model continuously improves with the increase of tasks." Should this conclusion consider which learning tasks to include?

**Suitability:**

2

---

### Meta-Review · Area_Chair_ifVF · 2024-07-03

**Recommendation:** Accept (Poster)
**Confidence:** 4

**Metareview:**

This paper got 1 weak accept, 3 borderline accept, and 2 borderline reject.  The Pros and Cons are as follows:

Pros:
1. This paper is well-organized and easy to follow.
2. The idea of progressive learning of multiple proxy tasks is novel in video anomaly detection (VAD).
2. The experiments demonstrate the efficacy of progressive learning in VAD.

Cons:
1. The motivation is not sufficient sufficiently supported which needs to further clarified.
2. The training paradigm is overall complex and the theriacal justification of the proposed method is weak.
3. Numerous writing.

After reading the rebuttals and the paper, I recommend an accept based on the novelty of the proposed method and holds potential for applications other than anomaly detection.